# The Role of the Compressor Isentropic Efficiency in Non-Intrusive Refrigerant Side Characterization of Transcritical CO$_2$ Heat Pump Water Heaters

**Francisco B. Lamas *** 🆔 **and Vítor A. F. Costa** 🆔

TEMA—Centre for Mechanical Technology and Automation, Department of Mechanical Engineering, Campus Universitário de Santiago, University of Aveiro, 3810-193 Aveiro, Portugal
* Correspondence: francisco.lamas@ua.pt

**Abstract:** Characterizing the refrigerant side of heat pump water heaters (HPWHs) can be intrusive and expensive. On the other hand, direct external measurement techniques can be unfeasible, particularly in commercial HPWHs for residential applications. Non-intrusive in situ characterization methods have already been successfully implemented in subcritical heat pumps. They provide the refrigerant mass flowrate and the equipment energy performance, by using contact temperature sensors and electric power meters. Subcritical suction and discharge-specific enthalpies necessary to apply the method can be obtained from the measured temperatures and their corresponding saturation pressures. Nevertheless, this approach does not apply to the transcritical CO$_2$ HPWHs. In the supercritical region, temperature and pressure are independent variables, and an iterative process regarding the compressor isentropic efficiency has to be considered. However, when isentropic efficiency data are not available, an additional procedure is required, using a validated gas cooler model to verify the physical reliability of the numerical solutions. This work aims at presenting base thermodynamic analysis of a novel methodology for non-intrusive refrigerant side characterization of transcritical CO$_2$ HPWHs, exploring the influence of the compressor isentropic efficiency condition.

**Keywords:** transcritical CO$_2$; heat pump water heater; compressor isentropic efficiency; non-intrusive characterization; gas cooler model





## 1. Introduction

Switching heating systems from fossil fuels to low-carbon alternatives is paramount for reaching the European climate objectives for 2030 and carbon neutrality by 2050. Heat pumps assume a primary function in accomplishing these targets, using energy from renewable sources (air, water, or geothermal), and being (mostly) electrically supplied and energy efficient, and thus contributing to a competitive, secure, and low-carbon economy [1]. Nevertheless, energy performance and low-carbon or renewable energy sources are not the only issues dictating their environmental impact. Heat pump technology is predominantly based on vapor-compression refrigeration systems, as in the current air-conditioning and refrigeration technologies. The commonly used refrigerants may substantially contribute to greenhouse gas emissions, particularly the fluorinated-based ones (F-gases) [2]. Atmospheric emissions during F-gas production, and leakages during operation, or even along the recovering, recycling, or destruction processes trigger the relaunch of some natural refrigerants, among them CO$_2$. Its environmental harmlessness, safety, low cost, high availability, and unique thermodynamic properties give this ultra-low global warming potential (GWP) operating fluid a significant advantage over other refrigerants [3]. Owing to its low critical temperature (31.1 °C), CO$_2$ is mainly used in transcritical vapor-compression cycles. One of the most widespread applications is the transcritical CO$_2$ heat pump water heater (TCO$_2$ HPWH) for residential applications,

particularly in Japan, where it is known as 'Eco Cute' and rated according to the Japanese energy efficiency standards.

In Europe, the energy performance of electrically driven HPWHs is rated according to the EN16147 standard [4]. However, the energy-performance indicator is used for equipment comparison and does not characterize the actual behavior of the HPWHs under a wide range of environmental conditions. On the other hand, it is based on waterside measurements and cannot provide any information for the refrigerant side, commonly obtained with intrusive and expensive equipment [5], unfeasible for in-situ measurements [6].

Non-intrusive methodologies applied in air-to-air heat pumps, based on compressor energy conservation (CEC), have demonstrated good accuracy [5,6]. Simplicity, reliability, independence, and non-interference in the system's operation are other advantages compared to indoor and outdoor air enthalpy-difference methods [6]. The CEC method allows an accurate estimation of the refrigerant mass flowrate and the equipment energy performance merely using (external) contact temperature sensors and electricity power meters [5,6]. The subcritical suction and discharge specific enthalpies necessary to apply the method are obtained from the measured temperatures and their corresponding saturation pressures. However, this method does not apply to the $TCO_2$ HPWHs. In the supercritical region, where both compressor discharge and gas cooler operating conditions fall, temperature and pressure are variables independent from each other, and an additional parameter or condition has to be considered—in this case, the compressor isentropic efficiency. It is worth mentioning that no in situ or non-intrusive methods for refrigerant-side characterization for transcritical $CO_2$ cycles have been found in the literature.

This work explores the role of the compressor isentropic efficiency in the non-intrusive refrigerant side characterization of $TCO_2$ HPWHs. The base thermodynamic analysis is presented, and this includes three versions regarding the compressor isentropic efficiency condition: first, constant; second, depending on the pressure ratio through an already known polynomial correlation; third, unknown. For the last version, an additional methodology is proposed and discussed, based on a validated model for the gas cooler energy balance. It allows for obtaining the discharge pressure and determining the compressor isentropic efficiency, thus, enabling non-intrusive HPWH refrigerant side characterization.

## 2. Materials and Methods

Figure 1 exhibits the schematic representation of a $TCO_2$ HPWH and the respective thermodynamic cycle on the $P - h$ and $T - s$ diagrams. In the $TCO_2$ HPWH scheme, the measurement equipment are also shown: 10 non-intrusive (external) contact temperature sensors for both water and refrigerant loops (2 and 8, respectively), one water mass flow meter (easily integrated in the water loop), and one electrical energy/power meter for the entire HPWH. The measurement outputs and variables considered in the following analysis are numbered according to the measurement devices represented in the figure. Note that measurement point 4 is irrelevant for the supercritical gas cooling characterization, yet crucial for an eventual condensation, as it provides the saturation temperature, similarly to point 8 (or point 7) for the evaporation process.

The $TCO_2$ HPWH operates in a transcritical cycle around the critical point, as illustrated in the $P - h$ diagram in Figure 1. Like any heat pump, it absorbs heat from a low-temperature heat source in the subcritical region (at the evaporator; evolution from point 7 to point 1 in Figure 1). The heat is rejected at a high-temperature heat source above the critical point, in the supercritical region (at the gas cooler; evolution from point 3 to point 5 in Figure 1). Unlike conventional (subcritical) systems, no condensation occurs, it being impossible to relate the corresponding (saturation) pressure and temperature. The supercritical $CO_2$ gas is cooled with a certain temperature glide, perfectly matching the water heating temperature profile (as depicted in the $T - s$ diagram, Figure 1, from "In" to "Out"). Owing to the high irreversibility losses during the expansion, $TCO_2$ HPWHs are equipped with a suction line heat exchanger (SLHX), enhancing the cycle performance. At

the SLHX, $CO_2$ is subcooled (point 5 to point 6 in Figure 1) before being throttled to the low-pressure level (point 6 to point 7 in Figure 1). The recovered heat from the subcooling process is transferred to the low pressure and temperature side (point 1 to point 2 in Figure 1) before the compression process (evolution from point 2 to point 3 in Figure 1).

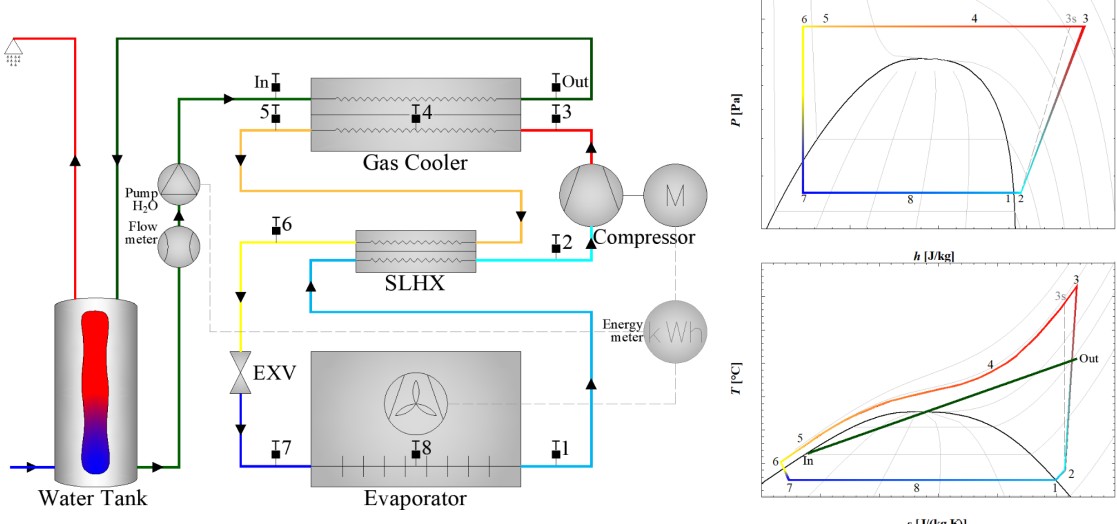

**Figure 1.** TCO$_2$ HPWH: scheme (**left**), and $P - h$ (**top left**) and $T - s$ (**bottom right**) diagrams.

The coefficient of performance of the whole TCO$_2$ HPWH is given by the ratio of heat transfer rate in the gas cooler, $\dot{Q}_{gasc}$ [W], to the total electrical input, $\dot{P}_{elec}$ [W],

$$COP = \dot{Q}_{gasc} / \dot{P}_{elec},\tag{1}$$

where the total electrical power input (with the compressor contribution, $\dot{P}_{elec_{comp}}$ [W], prevailing over the other active components, namely, the evaporator fan, water pump and other equipment such as control units, etc.) is

$$\dot{P}_{elec} = \dot{P}_{elec_{comp}} + \dot{P}_{elec_{fan}} + \dot{P}_{elec_{pump}} + \dot{P}_{elec_{others}} \text{ [W]}.\tag{2}$$

Neglecting the heat conduction along the tubes' walls and both the convective and radiant heat losses to the surroundings (owing to the good thermal insulation commonly used in its external envelope), the energy balance for the gas cooler can be written, for either the water or the refrigerant side, as

$$\begin{cases} \dot{Q}_{gasc} = \dot{m}_{H_2O} \bar{c}_{H_2O} \Delta T_{H_2O,\,gasc} \\ \dot{Q}_{gasc} = \dot{m}_{CO_2} \Delta h_{CO_2,\,gasc} \end{cases} \text{ [W]}.\tag{3}$$

From the previous system of equations, the refrigerant mass flowrate is given by

$$\dot{m}_{CO_2} = (\dot{m}_{H_2O} \bar{c}_{H_2O} \Delta T_{H_2O,\,gasc}) / \Delta h_{CO_2,\,gasc} \text{ [kg·s}^{-1}],\tag{4}$$

where all the waterside variables can be measured (mass flowrate, $\dot{m}_{H_2O}$ [kg·s$^{-1}$] and temperature increase, $\Delta T_{H_2O,\,gasc} = T_{out} - T_{in}$ [°C]), or calculated (specific heat, $\bar{c}_{H_2O}$ [J·kg$^{-1}$·K$^{-1}$]). By opposition, the specific enthalpy change on the refrigerant side ($\Delta h_{CO_2,\,gasc} = h_3 - h_5$ [J·kg$^{-1}$]) is unknown and depends on the $CO_2$ conditions at the gas cooler inlet and outlet. It becomes clear that the only way to obtain the refrigerant mass flowrate (without measuring it) is by determining both refrigerant specific enthalpies, $h_3$ and $h_5$ [J·kg$^{-1}$].

Using binary functions for the refrigerant properties (non-italic bold), the specific enthalpy results as a function of pressure (*P*) and temperature (*T*). Thus, for the gas cooler inlet and outlet, respectively,

$$h_3 = \mathbf{h}(P_3, T_3) \ [\text{J·kg}^{-1}], \tag{5}$$

$$h_5 = \mathbf{h}(P_5, T_5) \ [\text{J·kg}^{-1}]. \tag{6}$$

Disregarding the pressure drop in the gas cooler, pressure can be considered constant along its length. Therefore, with $P_5 = P_3$ [Pa], $h_5$ [J·kg$^{-1}$] also becomes dependent on $P_3$ [Pa]:

$$h_5 = \mathbf{h}(P_3, T_5) \ [\text{J·kg}^{-1}]. \tag{7}$$

Finally, the refrigerant mass flowrate, Equation (4), depends on only one unknown variable, $P_3$ [Pa], since both refrigerant temperatures, $T_3$ and $T_5$ [°C], and the respective specific enthalpies for that pressure can be obtained (likewise the waterside variables above mentioned).

$$\dot{m}_{CO_2} = [\dot{m}_{H_2O}\bar{c}_{H_2O}(T_{out} - T_{in})]/[\mathbf{h}(P_3, T_3) - \mathbf{h}(P_3, T_5)] \ [\text{kg·s}^{-1}]. \tag{8}$$

From the compressor isentropic efficiency definition,

$$\eta_{is} = (h_{3s} - h_2)/(h_3 - h_2). \tag{9}$$

Again, using binary functions for defining $h_{3s}$ [J·kg$^{-1}$], the specific enthalpy at the compressor discharge/gas cooler inlet (point 3 s) for the isentropic (ideal) compression, and $h_2$ [J·kg$^{-1}$], the specific enthalpy at the compressor suction/SLHX outlet (point 2), similarly to Equations (5) and (6), and rearranging Equation (9), it can be expressed as

$$\mathbf{h}(P_3, T_3) = \mathbf{h}(P_2, T_2) + [\mathbf{h}(P_{3s}, T_{3s}) - \mathbf{h}(P_2, T_2)]/\eta_{is} \ [\text{J·kg}^{-1}]. \tag{10}$$

Considering the non-pressure drop assumption, in both the high- and low-pressure sides, and the saturation pressure function (non-italic bold), results in the system of equations

$$\begin{cases} P_{3s} = P_3 \\ P_2 = \mathbf{P_{sat}}(T_8) \end{cases} [\text{Pa}], \tag{11}$$

where $T_8$ [°C] is the evaporation temperature.

The discharge specific entropy and temperature corresponding to the ideal compression can be written as

$$s_{3s} = s_2 = \mathbf{s}(P_2, T_2) \ [\text{J·kg}^{-1}\text{·K}^{-1}], \tag{12}$$

$$T_{3s} = \boldsymbol{T}(P_{3s}, s_{3s}) = \boldsymbol{T}(P_{3s}, s_2) = \boldsymbol{T}(P_{3s}, \boldsymbol{s}(P_2, T_2)) \ [°C] \tag{13}$$

or, through Equation (11), respectively, as

$$s_{3s} = \mathbf{s}(\mathbf{P_{sat}}(T_8), T_2) \ [\text{J·kg}^{-1}\text{·K}^{-1}], \tag{14}$$

$$T_{3s} = \mathbf{T}(P_3, \mathbf{s}(\mathbf{P_{sat}}(T_8), T_2)) \ [°C]. \tag{15}$$

Applying Equations (11), (14), and (15) in Equation (10),

$$\mathbf{h}(P_3, T_3) = \mathbf{h}(\mathbf{P_{sat}}(T_8), T_2) + \lceil \mathbf{h}(P_3, \mathbf{T}(P_3, \mathbf{s}(\mathbf{P_{sat}}(T_8), T_2))) - \mathbf{h}(\mathbf{P_{sat}}(T_8), T_2)\rceil/\eta_{is} \\ [\text{J·kg}^{-1}]. \tag{16}$$

At this point, three conditions can be considered regarding the compressor isentropic efficiency, each of them described in the following sections. The method, valid for the steady-state regime, should be applied for each working condition. A flowchart illustrating the methodology is presented in Figure 2.

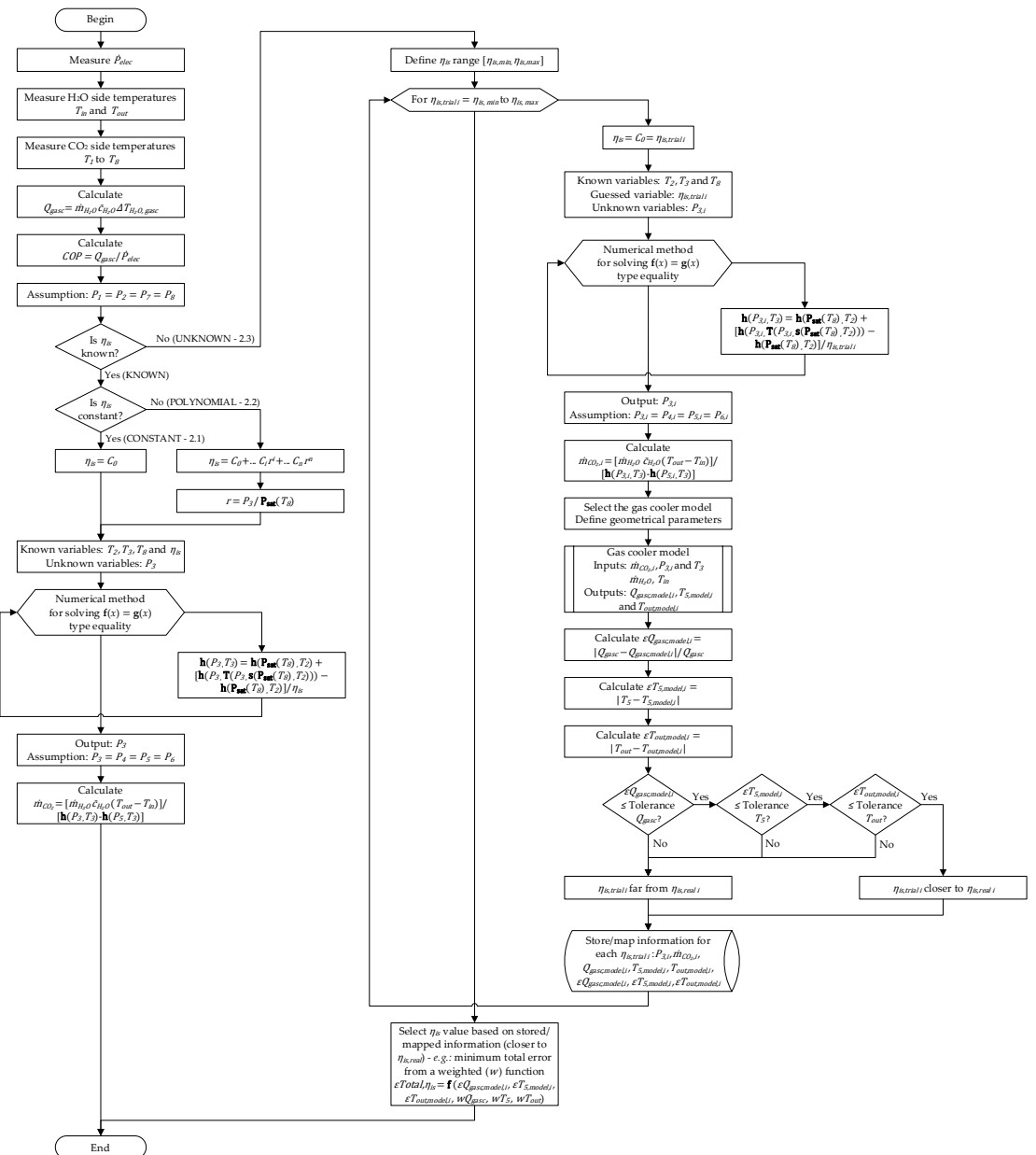

**Figure 2.** Flowchart of the proposed methodology.

### 2.1. Compressor Isentropic Efficiency Known as a Constant

Regarding the condition

$$\eta_{is} = C_0, \tag{17}$$

where $C_0$ is a known and constant value, Equation (16) can be written as the equality of two properties functions, $\mathbf{f}(x)$ and $\mathbf{g}(x)$, $\forall\, x \in \mathbb{R}^+$, each of them depending only on $P_3$ [Pa], since all other variables, $T_2$, $T_3$ and $T_8$ [°C], are known:

$$\mathbf{f}(P_3) = \mathbf{g}(P_3). \tag{18}$$

The equality of the two properties functions in Equation (18) can be solved through an iterative process, providing $P_3$ [Pa]. Knowing $P_3$ value that satisfies, Equation (16), it is possible to obtain the specific enthalpy at the gas cooler inlet and outlet through Equations (5) and (7), respectively. Furthermore, it is possible to obtain the refrigerant mass flow rate from the gas cooler energy balance equation written as in Equation (8).

## *2.2. Compressor Isentropic Efficiency Given by a Polynomial Correlation*

Many polynomial correlations for the compressor isentropic efficiency, as functions of the pressure ratio, can be found in the open literature. The most common are fourth-order ($n = 4$) and linear ($n = 1$) correlations [7–9]. However, the polynomial order depends on the compressor type, information provided by the compressor manufacturer, or on the regression analysis performed by the researchers. Regardless, the isentropic efficiency can be represented as

$$\eta_{is} = C_0 + \cdots + C_i r^i + \cdots + C_n r^n. \tag{19}$$

Each $C_i$ with $i \in \{0, 1, \ldots, n\}$ is a known and constant empirical value, and $r$ is the pressure ratio, which, combined with Equation (11) results in

$$r = P_3/P_2 = P_3/\mathbf{P_{sat}}(T_8) \tag{20}$$

Once more, in this case Equation (16) can also be written as the equality of two properties functions, $\mathbf{f}(x)$ and $\mathbf{g}(x)$, each of them depending only on $P_3$ [Pa]. In this case, the process for obtaining $P_3$ [Pa] is identical to that when the isentropic efficiency is given by a constant, as described in Section 2.1.

## *2.3. Unknown Compressor Isentropic Efficiency*

Compressor efficiency indicators and performance maps are commonly sensitive proprietary information, and therefore are often inaccessible. For this case, an iterative procedure is needed using a validated numerical model for the gas cooler energy balance. Many dimensional parameters and numerical and/or experimental data are available in the open literature. The information varies according to the system purpose (water heating, air conditioning, or refrigeration) and the gas cooler configuration, namely the single tube-in-tube [10] and multi-tubes-in-tube (straight [11] or twisted [12]), microchannel [13], brazed plate [14] or finned-tube [15]. For the TCO$_2$ HPWH, the most used configuration is the single tube-in-tube gas cooler, and the numerical model is commonly based on the finite volume method, using the logarithmic mean temperature difference approach [3]. The usual outputs of the gas cooler model are the heat transfer rate and the outlet temperatures for water and CO$_2$ ($\dot{Q}_{gasc}$ [W], $T_{out}$ and $T_5$ [°C], respectively); on the other hand, the main inputs are both the water mass flowrate and inlet temperature (respectively, $\dot{m}_{H_2O}$ [kg·s$^{-1}$] and $T_{in}$ [°C]), and the refrigerant mass flowrate and its inlet temperature and pressure ($\dot{m}_{CO_2}$ [kg·s$^{-1}$], $T_3$ [°C], and $P_3$ [Pa], respectively) [3,10–12,14]. Almost all these variables can be obtained, except $P_3$ [Pa] and $\dot{m}_{CO_2}$ [kg·s$^{-1}$] (which depends on the only unknown variable $P_3$ [Pa], as previously seen).

The non-measured input variables of the gas cooler model ($P_3$ [Pa] and $\dot{m}_{CO_2}$ [kg·s$^{-1}$]) are obtained through the process described in Section 2.1, attributing, in each iteration, a value for $C_0$ in Equation (17). The process will "sweep" a predefined isentropic efficiency range, providing data sets for the numerical simulation of the gas cooler model. A targeted definition of this range, decreasing the search field, can substantially improve the procedure efficiency by reducing the required computational time. As the lower limit, the compressor isentropic efficiency leading to the minimum pinch-point temperature difference between CO$_2$ and water temperature profiles, $\eta_{is\,min}$, is proposed, as represented on the left-hand side of Figure 3. The higher limit, $\eta_{is\,max}$, can be defined, at the most, as the unattainable isentropic (ideal) compression, depicted in Figure 3 (right-hand side).

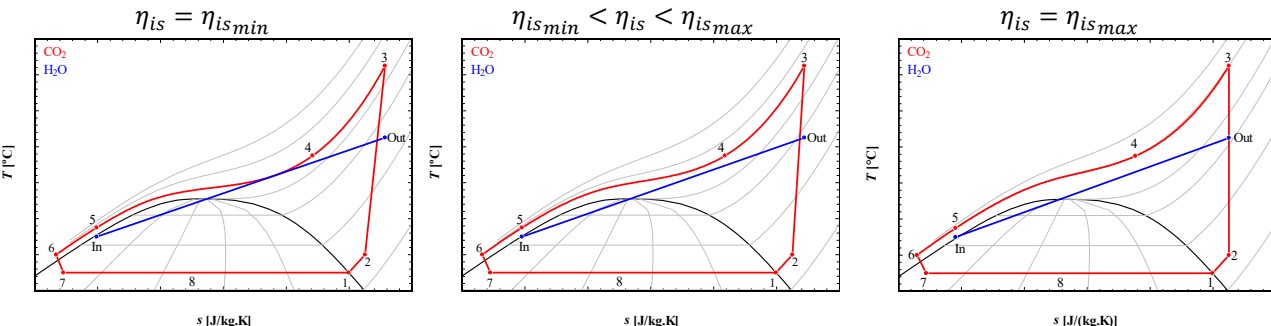

**Figure 3.** Definition of the search range for $\eta_{is}$.

However, enlarging the search range increases the computational time. For this reason, $\eta_{is_{max}}$ should be defined as the maximum known technical isentropic efficiency for the specific compressor type under consideration. Although, without having this information, it should be considered the maximum technical limit known at the date (around 0.9). The step between successive isentropic efficiencies trials can be adapted, or even refined, according to preliminary or previous results from wide-stepped iterations.

Finally, the validated gas cooler model is used to verify the physical reliability of the numerical solutions provided by the first procedure (defined in Section 2.1). As the second iteration process converges (i.e., the isentropic efficiency trial, $\eta_{is_{trial}}$, gets closer to the "real" value, $\eta_{is_{real}}$), the numerical results from the gas cooler model will approximate the experimental ones, as exhibited in Figure 4. While the three numerical outputs, $\dot{Q}_{gasc}$ [W], $T_{out}$ and $T_5$ [°C], are simultaneously within the respective and predefined acceptance tolerance, it is considered a valid $\eta_{is_{trial}}$. From the valid results ($\eta_{is_{trial}}$ close to $\eta_{is_{real}}$), it is possible to define a correlation for the isentropic efficiency with the form of Equation (19) based on the tested pressure ratios.

The results presented in Figure 4 are merely illustrative of the methodology's potential. They are a partial representation from the results of an experimental campaign on a commercial 4.5 kW TCO$_2$ HPWH (Sanden AquaECO2 GEU-45HPA+GEU-15QTA) regarding its energy performance characterization. Tests were conducted in a controlled climatic laboratory, performed based on the EN16147 standard [4] and extended for environmental conditions from −5 °C/75% to 25 °C/93%. The measurement devices are depicted in Figure 1, and their main characteristics are described in Table 1. The gas cooler model was developed and validated according to the methodology defined by Sánchez et al. [11] for the multi tubes-in-tube configuration.

**Table 1.** Measurement devices.

| Device | Type | Range/Calibration Range | Accuracy/Calibrated acc. |
|---|---|---|---|
| Thermocouples | K-type | −20~120 °C | ±1% rdg + 0.5 °C |
| Water flowmeter | Coriolis effect | 2~16 L/min | ±1% rdg + 0.12 L/min |
| Energy meter | Power quality analyzer | 120~400 V (@ 50 Hz) 0.1~5 A (@ 50 Hz) | ±0.1% rdg ±0.2% rdg |

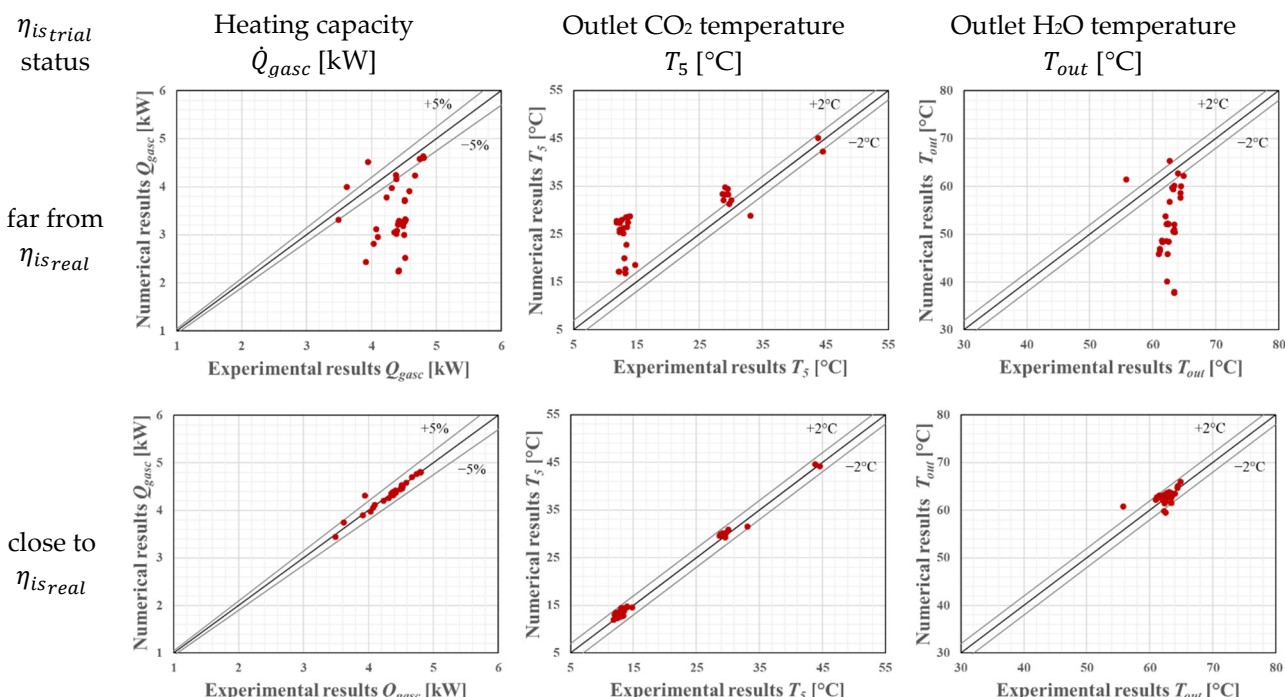

**Figure 4.** Example of the gas cooler model outputs for different $\eta_{is}$ trials and working conditions.

## 3. Conclusions

In this study, the thermodynamic basis for a novel non-intrusive refrigerant side characterization of transcritical $CO_2$ HPWHs is presented, evidencing the role of the compressor isentropic efficiency in the process. The complexity of the proposed methodology depends on knowledge about the compressor isentropic efficiency. When the compressor data are available, namely the compressor's isentropic efficiency, a simple iterative process is sufficient to obtain the discharge pressure since it is the only unknown required to close the equations system. However, when the compressor isentropic efficiency is unknown, another iterative process is required. A validated numerical model for the gas cooler energy balance is used to verify the physical reliability of the numerical solutions, searching over a range of possible candidates for the compressor isentropic efficiency. The proposed method can be similarly extended to transcritical $CO_2$ air conditioners and refrigeration systems, regardless of the type of gas cooler, widely filling the gap in non-intrusive and inexpensive refrigerant side characterization of ultra-low GWP vapor compression systems based on the transcritical $CO_2$ cycle.

**Author Contributions:** Conceptualization, methodology and analysis, F.B.L. and V.A.F.C.; writing—original draft preparation, F.B.L.; writing—review and editing, F.B.L. and V.A.F.C.; supervision, V.A.F.C. All authors have read and agreed to the published version of the manuscript.

**Funding:** This work was funded by the grant SFRH/BD/148378/2019 and the projects UIDB/00481/2020 and UIDP/00481/2020-FCT-Fundação para a Ciência e a Tecnologia; and CEN-TRO-01-0145-FEDER-022083-Centro Portugal Regional Operational Programme (Centro2020), under the PORTUGAL 2020 Partnership Agreement, through the European Regional Development Fund.

**Acknowledgments:** The present study was developed in the scope of the Smart Green Homes Project [PO-CI-01-0247-FEDER-007678], a co-promotion between Bosch Termotecnologia S.A. and the University of Aveiro. It is financed by Portugal 2020 under the Competitiveness and Internationalization Operational Program, and by the European Regional Development Fund.

**Conflicts of Interest:** The authors declare no conflict of interest.

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
