# Peer review of "The Role of the Compressor Isentropic Efficiency in Non-Intrusive Refrigerant Side Characterization of Transcritical CO2 Heat Pump Water Heaters"

_cleantechnol, doi:10.3390/cleantechnol4030050_

Round 1

Reviewer 1 Report

This manuscript describes a non-intrusive method to characterize the operation of a transcritical heat-pump water-heater using the minimum number of sensors without modifying the refrigerant-side cycle. Particularly, this manuscript uses non-immersion temperature sensors, an electrical power meter and a water flow meter for the secondary fluid. However, the method needs to develop and validate a well-contrasted gas-cooler model, which is difficult depending on the heat exchanger type used. Moreover, the validation of this last model requires additional measurement elements that have not been included in this methodology. Notwithstanding, it does not diminish the work developed by the authors, but it is important to remark on this aspect. Accordingly, two minor issues should be fixed in this manuscript:

- The gas-cooler model should be extended to remark the validation, operation range and characteristics (dimensions, length, diameter...)

- The use of a Coriolis mass flow meter in the secondary fluid (water) allows for minimizing the number of needed thermophysical properties (density), so, Why the authors do not use this parameter instead of volumetric flow?

Reviewer 2 Report

1.      T5 does not give the specific source, please specify the process of obtaining T5.

2.      Is relevant literature referred to in the derivation of Equations (1) - (16)? If so, please provide references.

3.      In this paper, when the correlation for the isentropic efficiency is known, the mass flow rate of CO2 can be obtained through a series of calculations. However, what is the purpose of the flow rate?

4.      The data points of ?̇???c and ???t are aggregated in Figure 3, but the data points of T5 are dispersed, please explain the reasons. In addition, do all data points fall within the respective and predefined acceptance tolerance? If not, is there a corresponding proportion? Please provide.

5.      In this paper, only some conclusions are given but data support is lacking. The experimental conditions in this paper are not clear, and experimental data under specific working conditions should be provided. Line 195, "From the results, it is possible to define a correlation for the isentropic efficiency, ". However, what is the relation of isentropic efficiency obtained through iteration?

6.      Do the results in this paper apply only to CO2 and H2O?

7.      The nomenclature "S-CO2" exists. However, does the nomenclature "TCO2" exist? Can it be used to refer to transcritical CO2?

8.      The abstract and the conclusion only provide the methods and conclusions, lacking data support. Moreover, the purpose of this paper is not clear, and the actual significance of isentropic efficiency derivation is not understood.

9.  Equations for enthalpy and entropy can be deleted. These values are easily obtained by REFPROP.

10.  There is no section of results and discussion.

Reviewer 3 Report

The paper proposed a novel method of non-intrusive refrigerant side characterization of transcritical CO2 HPWHs, and the role of the compressor isentropic efficiency in the process was presented. Several suggestions are listed as follows:

1.        Since the method proposed in the paper contains a series of calculations and programming, the flowchart of the whole calculation procedure is suggested to be presented.

2.        The error analysis of the simulation results and the experimental results is recommended.

3.        In section 2, it is better to give a brief description of how the transcritical CO2 HPWH system operates, which makes readers understand the contents more easily.

Round 2

Reviewer 2 Report

In the review of first-round, the reviewer has listed many issues. However, the authors don't agree with most of issues, and don't make revisions for the core issues. Although the authors have reponsed, but I really can't agree for point 8 ~point 10. If there is no results in a paper, it can't be accpeted for publication. For a methodology, the authors can apply for a patent, not suitable for publishing paper.  Therefore, I have to reject this paper.
